# Selective formation of acetate intermediate prolongs robust ethylene removal at 0 °C for 15 days

Mingyue Lin [1,2,3], Haifeng Wang[4], Takashi Takei[4], Hiroki Miura [3,4,5], Tetsuya Shishido [3,4,5], Yuhang Li [6], Jinneng Hu[6], Yusuke Inomata[7], Tamao Ishida[4], Masatake Haruta[4,9], Guangli Xiu[1] ✉ & Toru Murayama [3,8] ✉

Efficient ethylene ($C_2H_4$) removal below room temperatures, especially near 0 °C, is of great importance to suppress that the vegetables and fruits spoil during cold-chain transportation and storage. However, no catalysts have been developed to fulfill the longer-than-2-h $C_2H_4$ removal at this low temperature effectively. Here we prepare gold-platinum (Au-Pt) nanoalloy catalysts that show robust $C_2H_4$ (of 50 ppm) removal capacity at 0 °C for 15 days (360 h). We find, by virtue of *operando* Fourier transformed infrared spectroscopy and online temperature-programmed desorption equipped mass spectrometry, that the Au-Pt nanoalloys favor the formation of acetate from selective $C_2H_4$ oxidation. And this on-site-formed acetate intermediate would partially cover the catalyst surface at 0 °C, thus exposing active sites to prolong the continuous and effective $C_2H_4$ removal. We also demonstrate, by heat treatment, that the performance of the used catalysts will be fully recovered for at least two times.

Ethylene ($C_2H_4$) is a natural gaseous plant hormone and acts positively under controlled conditions as a ripening agent. However, even low parts-per-million (ppm) concentrations of $C_2H_4$ released from fresh products (such as fruits and vegetables) during shipping and storage would accelerate their deterioration at low temperatures (0−25 °C)[1], resulting in undesirable food waste[2–4]. Therefore, the elimination of $C_2H_4$ at a low temperature of ~0 °C is important for prolonging the shelf-life of food products[5].

Many catalysts and adsorbents for protecting $C_2H_4$-sensitive fresh products have been developed[6–12]. Unfortunately, all of them suffer the loss of $C_2H_4$ removal activity at 0 °C within 2 h. For example, the Pt/ mesoporous silica reported by Fukuoka et al. has shown an excellent capacity to remove $C_2H_4$ (less than 50 ppm) and has been commercially available since 2015[6]. However, this catalyst lost activity after 2 h-on-stream at 0 °C. Even though the activity of literature-reported catalysts could be mostly recovered after heat treatment, it is still imperative to develop robust and effective catalysts to long-term eliminate $C_2H_4$ at 0 °C for the storage and shipping of fresh products.

We found $C_2H_4$ favoring to convert into some intermediates (IMs) −such as acetic acid, acetaldehyde, ethanol, etc.−instead of final

[1]Shanghai Environmental Protection Key Laboratory on Environmental Standard and Risk Management of Chemical Pollutants, State Environmental Protection Key Laboratory of Environmental Risk Assessment and Control on Chemical Process, School of Resources and Environmental Engineering, East China University of Science and Technology, Shanghai 200237, PR China. [2]Shanghai Institute of Pollution Control and Ecological Security, Shanghai 200092, PR China. [3]Research Center for Hydrogen Energy-based Society, Graduate School of Urban Environmental Sciences, Tokyo Metropolitan University, 1-1 Minami-Osawa, Hachioji, Tokyo 192-0397, Japan. [4]Department of Applied Chemistry for Environment, Graduate School of Urban Environmental Sciences, Tokyo Metropolitan University, 1-1 Minami-Osawa, Hachioji, Tokyo 192-0397, Japan. [5]Elements Strategy Initiative for Catalysts & Batteries, Kyoto University, Kyoto 615-8520, Japan. [6]School of Materials Science and Engineering, East China University of Science and Technology, Shanghai 200237, PR China. [7]Faculty of Advanced Science and Technology, Kumamoto University, 2-39-1 Kurokami, Chuo-ku, Kumamoto 860-8555, Japan. [8]Yantai Key Laboratory of Gold Catalysis and Engineering, Shandong Applied Research Center of Gold Nanotechnology (Au-SDARC), School of Chemistry & Chemical Engineering, Yantai University, Yantai 264005, PR China. [9]Deceased: Masatake Haruta. ✉e-mail: xiugl@ecust.edu.cn; murayama@tmu.ac.jp

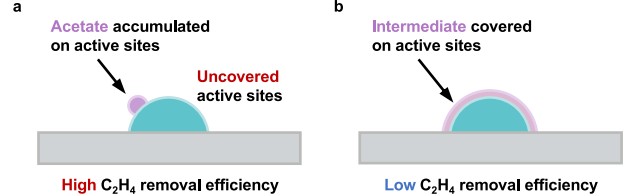

**Fig. 1 | Schematic diagram of C$_2$H$_4$ removal over catalysts with different conditions of intermediates. a** When generating solid-like acetate as intermediate which would accumulate on the surface of catalyst, the active sites will be uncovered and thus leading to the continued high-efficiency C$_2$H$_4$ removal. **b** When forming liquid-like intermediate on the surface of catalyst, the active sites will be covered and thus resulting in low C$_2$H$_4$ removal efficiency.

carbon dioxide (CO$_2$) on catalyst surface, especially at low temperatures[7,8]. We then noted, among these IMs, that acetic acid (AcOH) could be a suitable IM for eliminating C$_2$H$_4$ at 0 °C, because its solid-like feature at 0 °C (solidification temperature of 16.6 °C) will expose active sites for continuing C$_2$H$_4$ elimination (Fig. 1a). Meanwhile, if the C$_2$H$_4$ removal tests are carried out at room temperature (25 °C) when this on-site-formed acetic acid will be liquid-like that will spread on the catalyst surface (Fig. 1b), the catalyst may be quickly deactivated and thus losing the C$_2$H$_4$ removal activity. Therefore, we reasoned that a catalyst, which is able to selective form acetic acid during C$_2$H$_4$ eliminating, would exhibit robust C$_2$H$_4$ removal performance at 0 °C.

We should note that the choices of both catalysts and support materials are critical for C$_2$H$_4$ removal. Alloy nanoparticles (NPs) are important catalysts for the conversion of hydrocarbons[13] due to the different electronic states and structures compared to the monometallic NPs[14]. For example, the dominant adsorption mode of C$_2$H$_4$ on a Pt(111) surface is the di-σ adsorbed mode up to ~250 K[15–17], and it would change to a stable phase of ethylidyne species (CH$_3$C) as the temperature increased[18,19]. Introduction of other metals would result in geometric and electronic modulations on Pt and therefore optimize the key adsorption species on Pt, such as the addition of Sn would suppress the formation of ethylidyne[20] and the addition of Au would enhance the interaction between Pt and ethylidyne species[21]. In addition, C$_2$H$_4$ can adsorb on Au sites at 263 K and 203 K with a lower heat of adsorption than that on Pt sites[21]. Therefore, we expect that the Au-Pt alloy could be a good catalyst candidate for the adsorption and transformation of C$_2$H$_4$ at low temperatures. For the support materials, given that Brønsted acidity promotes the adsorption of C$_2$H$_4$[10–12], we select Mordenite 20 (denoted as ZHM20) as the acidic support for Au-Pt alloy NPs due to its abundant acid sites, especially Brønsted acid sites, and large surface area[9].

Motivated by this, here we develop gold-platinum (Au-Pt) alloy NPs as catalysts for the adsorption and selective transformation of C$_2$H$_4$ into AcOH at 0 °C, with ZHM20 as the support that could facilitate catalysts dispersion and enhance C$_2$H$_4$ adsorption. We show that the Au-Pt/ZHM20 catalysts present robust and efficient removal of 50 ppm C$_2$H$_4$ at 0 °C with a steady 80% efficiency for 40 h, which is ca. 30-fold longer than the best results in literature. We also find that the catalysts operate for 15 days to continuous eliminate 50 ppm C$_2$H$_4$ and the removal efficiency is fully recovered after heat treatment. In addition, we note a fast deactivation on the same catalysts within 5 h for C$_2$H$_4$ removal at 25 °C. We reveal, by a series of *operando* and online measurements, that the AcOH is *on-site* formed on the catalysts during C$_2$H$_4$ removal tests on Au-Pt/ZHM20, resulting the contrary stability at 0 and 25 °C.

## Results
### Structure analyses of catalysts
We deposited Au-Pt alloy on ZHM20 with a total metal loading amount ca. 1 wt% through a sol immobilization method (see experimental

details in "Methods"). We also prepared Au/ZHM20 and Pt/ZHM20 as controls (Supplementary Figs. 1 and 2). We defined the as-prepared Au-Pt as Au$_{54}$Pt$_{46}$ (molar ratio) based on the inductively coupled plasma atomic emission spectrometry (ICP-AES) results (Supplementary Table 1). We revealed the formation of bimetallic alloy of Au-Pt/ZHM20 according to the HAADF-STEM image and corresponding elemental mappings (Fig. 2a–d), showing the well dispersed Au and Pt elements in NPs. In the XRD pattern of Au$_{54}$Pt$_{46}$/ZHM20 (Supplementary Fig. 2), we observed no diffraction peaks ascribed to Au(111) or Pt(111) but a broad peak centered at 38.8°, suggesting the formation of Au-Pt alloy. We further calculated surface areas of Au$_{54}$Pt$_{46}$/ZHM20 together with other two controls and bare support ZHM20 (calcined at 500 °C) to be from 766 m$^2$ g$^{-1}$ to 835 m$^2$ g$^{-1}$ by nitrogen adsorption and desorption isotherms (Supplementary Fig. 3 and Supplementary Table 2). We found that both Au$_{54}$Pt$_{46}$/ZHM20 and bare support ZHM20 have strong Brønsted acidity by NH$_3$-TPD (Supplementary Fig. 4) and FT-IR of pyridine adsorption (Supplementary Fig. 5), which would favor the C$_2$H$_4$ adsorption[10–12]. Since the inner pore size of ZHM20 is only 0.58 nm (Supplementary Table 2), much smaller than the sizes of metal nanoparticles, we thus conclude that the metal particles are deposited on the exterior surface of the support. We also noted that the large particle size of Au was observed in Au/ZHM20 catalyst than Au-Pt alloy NPs in Au$_{54}$Pt$_{46}$/ZHM20, which is probably due to the fact that the sol immobilization method that might be unsuitable to deposit Au NPs compared to Pt and Au-Pt alloys. However, in order to compare the performance of these catalysts, we used the same preparation process in this work.

To in-depth investigate the electronic states of Au$_{54}$Pt$_{46}$/ZHM20, we conducted XAFS measurements, with the two controls and Au and Pt foils as references, by collecting Au $L_3$-edge (Fig. 2e) and Pt $L_3$-edge (Fig. 2f) XANES spectra. We noted that the shapes and absorption edge energies of the spectra of Au$_{54}$Pt$_{46}$/ZHM20 are close to those of references, suggesting that the Au$_{54}$Pt$_{46}$ is metallic. We magnified the graphs as insets to compare the white line intensities. We noticed a lower white line intensity of Au$_{54}$Pt$_{46}$/ZHM20 at 11921 eV in Au $L_3$-edge, and a higher white line intensity at 11562 eV in Pt $L_3$-edge. This reverse trend of white line intensities indicates that the charge transfer from Pt to Au occurred after alloying, forming the electron-rich Au species and electron-deficient Pt in the Au$_{54}$Pt$_{46}$ NPs[14,22–24]. The addition of Au into Pt could lead to attractive interaction between Pt and ethylidyne species[21], which may facilitate the catalytic conversion of C$_2$H$_4$.

### C$_2$H$_4$ removal performance of Au$_{54}$Pt$_{46}$/ZHM20
We performed C$_2$H$_4$ removal tests at 0 °C controlled by using an ice bath under 50 ppm C$_2$H$_4$/20%O$_2$/N$_2$ with a total flow rate of 10 mL min$^{-1}$ (see details in "Methods"). We noted a U-shaped C$_2$H$_4$ removal efficiency curve with a turning point at ca. 3.5 h on Au$_{54}$Pt$_{46}$/ZHM20 catalyst as shown in Fig. 3a. This U-shaped curve could be originated from the overlap of two curves: one is the C$_2$H$_4$ adsorption curve (like the black curve of the bare ZHM20 support in Fig. 3a) and the other is the C$_2$H$_4$ catalytic converting curve. We noticed that the catalyst may need to adsorb a minimum amount of C$_2$H$_4$ before the reaction is initiated. This is because the support contains abundant acid sites, especially Brønsted acid sites that may more favor the C$_2$H$_4$ adsorption than Au-Pt alloys. Therefore, the catalytic reaction for selectively converting C$_2$H$_4$ could not be started owing to the lack of C$_2$H$_4$ reactant on Au-Pt alloy catalysts, since most of C$_2$H$_4$ molecules would be trapped by the ZHM20 support in the initial stage. In the steady state after 3.5 h, this catalyst presents a high C$_2$H$_4$ removal efficiency (>80%) for at least 40 h. This reaction period is the first demonstration of long-term and efficient C$_2$H$_4$ removal, which is more than 30 times higher than the best catalysts operated at 0 °C in the literatures (Fig. 3b and Supplementary Table 3)[4,6,10,12,25,26]. We calculated the C$_2$H$_4$ removal rate on Au$_{54}$Pt$_{46}$/ZHM20 in the steady state at 0 °C to be 120 mL$_{(ethylene)}$/kg h, which is ~5× higher than the reported commercially used Pt/SBA-15

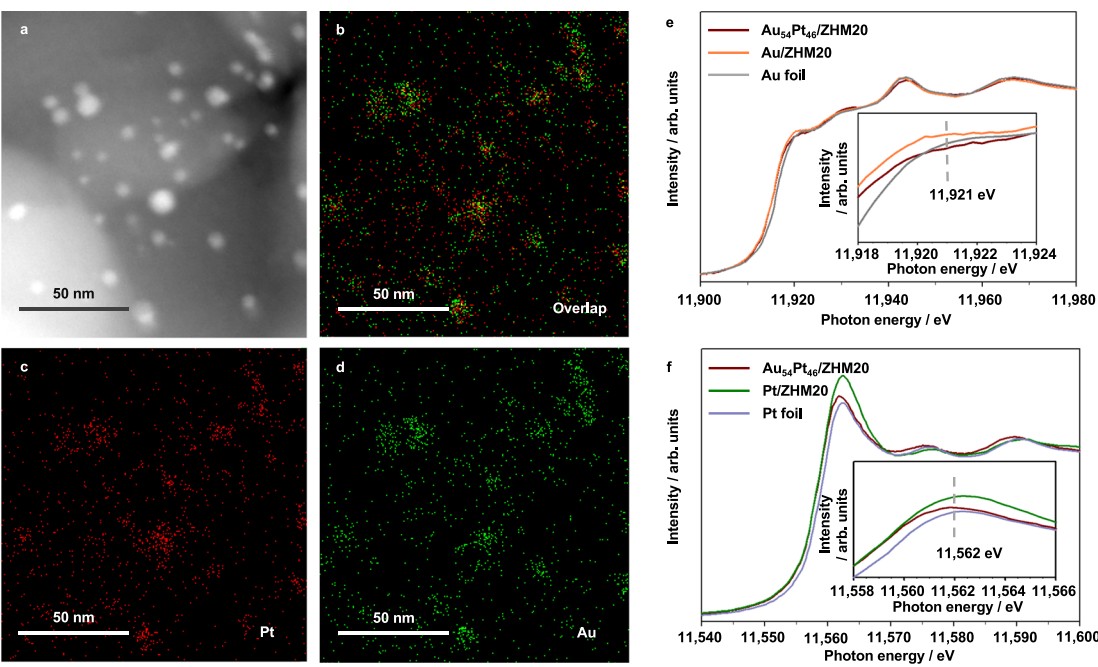

**Fig. 2 | Structure analyses of Au₅₄Pt₄₆/ZHM20. a** HAADF-STEM image, **b**–**d** corresponding elemental mappings, **e** Au $L_3$-edge and **f** Pt $L_3$-edge XANES spectra (Insets show magnifications around the white lines) of Au₅₄Pt₄₆/ZHM20. Units arbitrary units.

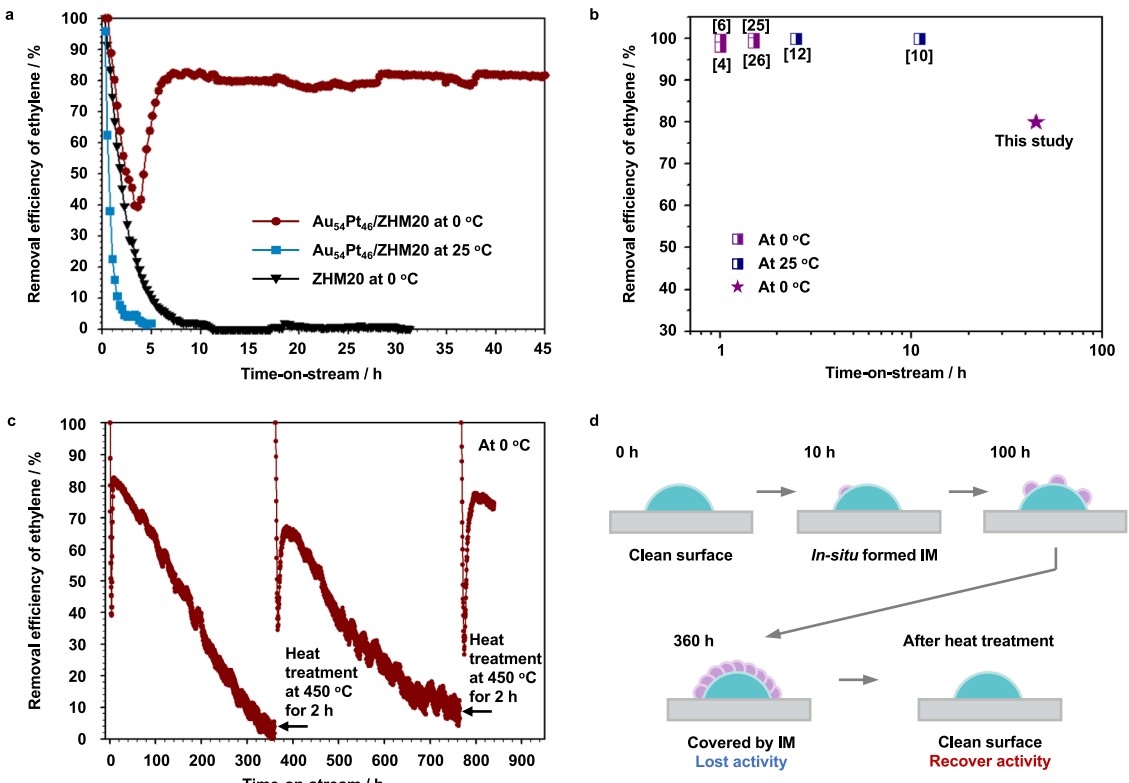

**Fig. 3 | C₂H₄ removal performance. a** C₂H₄ removal efficiencies with time-on-stream over ZHM20 and Au₅₄Pt₄₆/ZHM20 at 0 °C or 25 °C (reaction condition: 50 ppm C₂H₄, 20% O₂ and N₂ balance; catalyst, 0.2 g; space velocity, 3000 mL h⁻¹ g⁻¹). **b** C₂H₄ removal efficiency and stability over Au₅₄Pt₄₆/ZHM20 in comparison with recent reports[4,6,10,12,25,26]. **c** Time courses for C₂H₄ removal over Au₅₄Pt₄₆/ZHM20 at 0 °C. Heat treatment was conducted at 450 °C for 2 h under N₂ flow (50 mL min⁻¹). **d** Schematic diagram of the deactivation and recovery processes of Au₅₄Pt₄₆/ZHM20.

(25 mL$_{(ethylene)}$/kg × h)[7]. This rate is also much higher than that of C₂H₄ generated by fruits, such as apple (0.28 mL$_{(ethylene)}$/kg h) according to the semi-practical conditions for the preservation of perishables[27], proving the promising application possibility.

We continued to examine the C₂H₄ removal stability at 0 °C of Au₅₄Pt₄₆/ZHM20 (Fig. 3c). We took as long as 15 days (360 h) that the removal efficiency gradually decreased from 80% to 0% for continuous removing C₂H₄ with a total removed quantity of 4.4 mL. We recovered

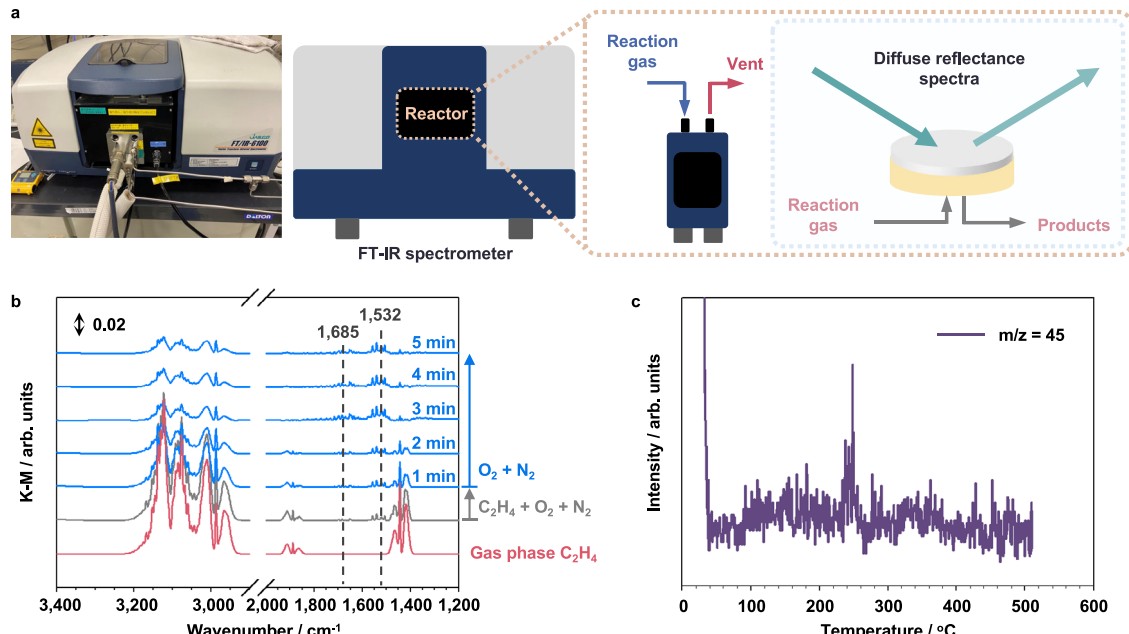

**Fig. 4 | IMs investigations during C$_2$H$_4$ removal. a** Schematic diagram of the DRIFT spectroscopy measurement. **b** DRIFT spectra of C$_2$H$_4$ oxidation over Au$_{54}$Pt$_{46}$/ZHM20 at 0 °C. The sample was pretreated under N$_2$ flow (50 mL min$^{-1}$) at 250 °C for 1 h. After cooling to 0 °C, the background spectrum was taken under N$_2$ flow. Then a mixture of C$_2$H$_4$ (25 mL min$^{-1}$), O$_2$ (20 mL min$^{-1}$), and N$_2$ (55 mL min$^{-1}$) was flowed for 30 min, and the flow of C$_2$H$_4$ was stopped while keeping the flow of

O$_2$ and N$_2$ for 5 min. **c** TPD profile of acetic acid of the used Au$_{54}$Pt$_{46}$/ZHM20. Reaction conditions: C$_2$H$_4$ oxidation was carried out on Au$_{54}$Pt$_{46}$/ZHM20 (0.2 g) at 0 °C for 10 h (81% conversion), and then the used Au$_{54}$Pt$_{46}$/ZHM20 (0.1 g) was transferred to measure TPD under He flow (30 mL min$^{-1}$) from 25 °C to 500 °C at a ramp rate of 5 °C min$^{-1}$. During the desorption, the mass signals of possible products were recorded. Units arbitrary units.

the excellent removal efficiency (>80%) of the spent Au$_{54}$Pt$_{46}$/ZHM20 via heat treatment at 450 °C for 2 h under N$_2$ flow. We then demonstrated the re-treated Au$_{54}$Pt$_{46}$/ZHM20 exhibiting robust C$_2$H$_4$ removal efficiency at 0 °C for the other 15 days, same as the fresh one. Even after the second-run heat treatment of the spent Au$_{54}$Pt$_{46}$/ZHM20, the initial removal efficiency recovered to 100% and was maintained at >75% in the steady state for 40 h. We thus propose the possible deactivation and recovery processes of Au$_{54}$Pt$_{46}$/ZHM20 as illustrated in Fig. 3d. In details, the *on-site* formed solid-like IMs (such as AcOH at 0 °C) will continually accumulate on surface and cover the active sites of the catalysts, leading to the gradually decreased C$_2$H$_4$ removal efficiency. After the active sites are fully covered, the catalysts will lose the activity for eliminating C$_2$H$_4$. The heat treatment of the used catalysts will clean the IMs accumulated on surface and thus the initial removal efficiency will be recovered. Although the heat treatment will make it difficult to incorporate the catalyst into food packaging materials, we expect the usage of this catalyst in a box-like device with air flow system, which will locate in the space for cold-chain storage and transportation.

When we increased the reaction temperature to 25 °C, we found a quick deactivation on Au$_{54}$Pt$_{46}$/ZHM20−from 100% to 0% of C$_2$H$_4$ removal efficiency−within 5 h for the reaction (Fig. 3a), which is contrary behavior compared to that at 0 °C. We therefore conducted *operando* time-dependent diffuse reflectance infrared Fourier transform (DRIFT) spectroscopy measurement (Fig. 4a) under the conditions of 25%C$_2$H$_4$/20%O$_2$/N$_2$ with a flow rate of 100 mL min$^{-1}$ at 0 °C. DRIFT spectra of the C$_2$H$_4$ removal process on Au$_{54}$Pt$_{46}$/ZHM20 are shown in Fig. 4b. The infrared spectrum of gas-phase C$_2$H$_4$ is provided as background, and the bands for C$_2$H$_4$ locate in three regions: 3200−2900 cm$^{-1}$, 1900−1800 cm$^{-1}$, and 1500−1400 cm$^{-1}$[28,29]. The bottom dark gray line is the infrared spectrum under a mixture flow of C$_2$H$_4$/O$_2$/N$_2$ at 0 °C. To rule out the possible overlap between the bands of IM products and the gas-phase C$_2$H$_4$ peaks, we stopped C$_2$H$_4$ flow after 30 min and continued flowing the mixture of O$_2$/N$_2$. The absorption bands at $\tilde{v}$ = 1532 cm$^{-1}$ on Au$_{54}$Pt$_{46}$/ZHM20 correspond to the antisymmetrical stretching vibration of surface carboxylates, an

acetate-based IM such as AcOH[30,31]. The absorption bands centered at $\tilde{v}$ = 1685 cm$^{-1}$ assigned to C=O stretching[32,33] also suggest the possible existence of AcOH. While the broad bands around $\tilde{v}$ = 1650 cm$^{-1}$ could be assigned to the adsorbed H$_2$O[34]. We should note that the intensity of these bands for AcOH enhanced while those for C$_2$H$_4$ decreased with increasing time, indicating the selective oxidation of C$_2$H$_4$ into AcOH on Au$_{54}$Pt$_{46}$/ZHM20.

For comparison, we also conducted the DRIFT measurements on Au/ZHM20 and Pt/ZHM20. For Pt/ZHM20 (Supplementary Fig. 6a), we observed the intensities of gas-phase C$_2$H$_4$ bands vanished at 3 min after we stopped feeding C$_2$H$_4$; meanwhile, the bands assigned to C=O stretching at 1685 cm$^{-1}$ appeared at this point. This suggests the adsorbed C$_2$H$_4$ on Pt/ZHM20 converted to AcOH intermediate. However, the C$_2$H$_4$ removal performance of Pt/ZHM20 is only ~50% (Supplementary Fig. 6b), indicating that the vanished C$_2$H$_4$ on Pt/ZHM20 in DRIFT measurements are owing to the fast desorption as well as the conversion into AcOH. For Au/ZHM20 (Supplementary Fig. 6c), we found that the C$_2$H$_4$ bands remained at initial intensity while negligible signals for C=O stretching during the DRIFT tests after stopping the C$_2$H$_4$ feed. Together considering the modest C$_2$H$_4$ removal of ~50% of this catalyst (Supplementary Fig. 6d), we reasoned that the C$_2$H$_4$ would be strongly adsorbed on Au/ZHM20 but hard to convert into AcOH. Based on these results, we thus propose that the excellent performance of Au$_{54}$Pt$_{46}$/ZHM20 for removing C$_2$H$_4$ at 0 °C could be due to the suitable C$_2$H$_4$ adsorption ability and high catalytic activity of C$_2$H$_4$-to-AcOH conversion.

We also carried out the temperature-programmed desorption (TPD) equipped with online mass to detect the possible IMs or transformed species of C$_2$H$_4$ formed on Au$_{54}$Pt$_{46}$/ZHM20. We detected AcOH−a sharp peak indicating the desorption of AcOH at ~250 °C in the TPD profile (Fig. 4c)−along with C$_2$H$_4$ (unremoved) and water (Supplementary Fig. 7) in the downstream of the Au$_{54}$Pt$_{46}$/ZHM20 after its removal efficiency has reached the steady state at 0 °C for 10 h. Based on the above measurements, we noted, by possessing electron-deficient Pt and electron-rich Au, that Au$_{54}$Pt$_{46}$/ZHM20 may be

beneficial for selectively forming AcOH during the $C_2H_4$ removal. Therefore, when we consider the solidification temperature of AcOH is 16.6 °C, the AcOH IM would be accumulated on the surface of catalysts as a solid-like feature at the test temperature of 0 °C, thereby exposing active sites that fulfill the long-term and robust $C_2H_4$ removal (Fig. 1a). In contrast, at 25 °C, the *on-site* formed AcOH could be a liquid-like IM that would spread on surface and quickly cover all active sites, thus deactivating the catalysts (Fig. 1b). We also used molecular dynamics (MD) simulations to examine the interface force between AcOH and Au-Pt nanoalloy at different temperatures (Supplementary Fig. 8). We found that the binding force between AcOH molecules and the catalyst is stronger at a higher temperature (interface force of −621.5 kcal/mol at 298 K) than that at a lower temperature (interface force of −585.7 kcal/mol at 100 K). The strong binding force between AcOH and Au-Pt at the higher temperature would result in the AcOH spreading on the catalyst surface, while the weak binding force would make AcOH tend to agglomerate like solid. It is worth noting that, although the set temperatures in MD simulations are different compared to reality, the trends shown here consist of the above experimental results.

To rule out the possibility that the support may influence the $C_2H_4$ removal efficiency, we also performed the reactions at similar conditions using the bare support ZHM20. As shown in Fig. 3a, the initial $C_2H_4$ removal efficiency in the first 15 min on ZHM20 is 100%, and it reached the maximum adsorption capacity after flowing the feed gas for 11 h (total $C_2H_4$ adsorption capacity of 0.074 mmol g$^{-1}$). Although ZHM20 is a zeolite with a large amount of Brønsted acid sites that could be used for adsorbing $C_2H_4$ (3.5 mmol g$^{-1}$, Supplementary Fig. 9), it may favor adsorbing $O_2$ instead of $C_2H_4$ under the reaction conditions.

In order to further evaluate the durability of the $Au_{54}Pt_{46}$/ZHM20 catalyst developed in this work, we stored the catalyst for two years and heat-treated it once again at 450 °C for 2 h under $N_2$ flow to regenerate the catalyst. We found that the conversion efficiency of $C_2H_4$ removal can still achieve 75% (Supplementary Fig. 10), suggesting the excellent stability of the catalyst. We also investigated the performance under different $C_2H_4$ concentrations and flow rates (Supplementary Figs. 10 and 11). We noted, at a low $C_2H_4$ concentration of 25 ppm, that the catalyst exhibits a delay activation and a similar $C_2H_4$ removal efficiency compared to those of 50 ppm, suggesting the transport limitation under the condition of 25 ppm $C_2H_4$. However, when we increased the $C_2H_4$ concentration to 50 ppm or higher 100 ppm, the $C_2H_4$ concentrations and flow rates may have negligible influence on the $C_2H_4$ removal activity of $Au_{54}Pt_{46}$/ZHM20 catalyst (Supplementary Note 1).

## Comparison with controls for $C_2H_4$ removal

We prepared two more Au-Pt alloy NPs with different molar ratios of $Au_{15}Pt_{85}$ and $Au_{77}Pt_{23}$ (Supplementary Table 1 and Supplementary Figs. 12 and 13) to investigate whether the Au and Pt amounts will affect $C_2H_4$ removal performance. The XRD profiles of three Au-Pt/ZHM20 are shown in Supplementary Fig. 12. The HAADF-STEM images and size distributions of the three Au-Pt/ZHM20 catalysts are shown in Supplementary Fig. 13. The average sizes are 5.8 ± 2.0 nm, 6.5 ± 2.1 nm, and 8.4 ± 2.8 nm for $Au_{15}Pt_{85}$, $Au_{54}Pt_{46}$, and $Au_{77}Pt_{23}$, respectively. The HRTEM images of the Au-Pt alloy NPs containing clear fringe spacings (Supplementary Fig. 14) demonstrate their high crystalline feature. We also detected the elemental mappings of the controls (Supplementary Figs. 15 and 16), which reveals that Au and Pt can be homogeneously dispersed in NPs. We noticed that the introduction of Au into Au-Pt alloy NPs would increase the sizes of alloy NPs; however, all three Au-Pt/ZHM20 samples showed similar surface areas (802–826 m$^2$/g, Supplementary Fig. 17), pore sizes (0.58 nm, Supplementary Fig. 17), and acid amounts (0.96–1.0 mmol/g, Supplementary Fig. 18 and Supplementary Table 2). Additional Au $L_3$-edge and Pt $L_3$-edge XAFS

measurements suggest that all three Au-Pt/ZHM20 samples possessed electron-deficient Pt and electron-rich Au in nanoalloys (Fig. 5a and b).

Figure. 5c shows a comparison of $C_2H_4$ removal efficiencies at 0 °C over the three Au-Pt/ZHM20 catalysts together with solely Au or Pt loaded ones. Again, we observed U-shaped removal curves with similar removal efficiencies of 77%, 81%, and 83% in the steady state for $Au_{15}Pt_{85}$/ZHM20, $Au_{54}Pt_{46}$/ZHM20, and $Au_{77}Pt_{23}$/ZHM20, respectively. This suggests that the molar ratios of Au and Pt have negligible influence on the removal efficiency at 0 °C. The high $C_2H_4$ removal efficiency in the steady state lasted 33 h and 25 h for $Au_{15}Pt_{85}$/ZHM20 and $Au_{77}Pt_{23}$/ZHM20, respectively. Together considering the curve trends of Au/ZHM20 and Pt/ZHM20 controls, we found, at the middle molar ratio of Au/Pt, that the $Au_{54}Pt_{46}$ alloy NPs will facilitate the $C_2H_4$ removal, while higher or lower Au/Pt ratios show a closer performance to Au/ZHM20 or Pt/ZHM20, respectively.

We also summarized the steady $C_2H_4$ removal performance on the above catalysts under different temperatures (Fig. 5d). After 25 °C, we found that catalytic oxidation of $C_2H_4$ to $CO_2$ occurred and the efficiency for removal of $C_2H_4$ increased with an increase in the temperature (Supplementary Fig. 19). With a decrease in the ratio of Pt in the catalysts, the efficiency for catalytic removal of $C_2H_4$ and the corresponding yield of $CO_2$ decreased in the order of Pt/ZHM20 > $Au_{15}Pt_{85}$/ZHM20 > $Au_{54}Pt_{46}$/ZHM20 > $Au_{77}Pt_{23}$/ZHM20 > Au/ZHM20 as the temperature was increased above room temperature, suggesting that Pt NPs are more favorable than Au NPs for catalytic conversion of $C_2H_4$ to $CO_2$. The support ZHM20 also showed catalytic activity for the conversion of $C_2H_4$ to $CO_2$ at temperatures higher than 80 °C and the $CO_2$ yield reached 60% at 260 °C. Although the ZHM20 exhibits activity for $C_2H_4$ conversion at high temperatures, considering that the actual shipping and storage conditions of $C_2H_4$ released from fruits and vegetables are at low temperatures (0–5 °C), the high efficiency, long-term stability, and excellent recovery features of $Au_{54}Pt_{46}$/ZHM20 for $C_2H_4$ removal at 0 °C can make it a promising material for further practical use. Moreover, comparing this catalytic process with other existing solutions for eliminating $C_2H_4$, we noticed that most of the traditional $C_2H_4$ removal methods have shortcomings. For example, adsorbents such as activated carbon cannot be used for a long time due to the limited adsorption capacity; chemical oxidants are toxic and contain potential safety hazards during long-term use; photocatalytic technology requires high equipment costs because of the need for ultraviolet light sources. Therefore, the catalytic process, especially when we use a catalyst with robust activity and stability such as $Au_{54}Pt_{46}$/ZHM20 produced in this work, would provide new opportunities for removing the trace amount of $C_2H_4$ for a long time at low temperatures.

## Discussion

In summary, we developed a robust and effective $Au_{54}Pt_{46}$/ZHM20 catalyst for long-term eliminating trace $C_2H_4$ with a high removal efficiency of ~80% at 0 °C. This catalyst showed two stages of $C_2H_4$ removal: the first stage was attributed to the adsorption of $C_2H_4$ and the second could be due to the catalytic conversion of $C_2H_4$ to IMs on $Au_{54}Pt_{46}$/ZHM20. Investigations by *operando* DRIFTs and online TPD measurements suggested that AcOH as IM is on-site-formed on the catalyst during $C_2H_4$ elimination. A long-term stability test over $Au_{54}Pt_{46}$/ZHM20 demonstrated the usage for 15 days at 0 °C under continuous feed gas containing 50 ppm of $C_2H_4$. The used $Au_{54}Pt_{46}$/ZHM20 will be reactivated by heat treatment. The robust and long-term $C_2H_4$ removal ability of $Au_{54}Pt_{46}$/ZHM20 at 0 °C makes it an excellent candidate for $C_2H_4$-sensitive applications. This work may also provide new insight into designing robust catalysts for $C_2H_4$ removal by modulating the transformed $C_2H_4$ species at specific operating temperatures, such as on-site-forming AcOH that leads to long stability at 0 °C.

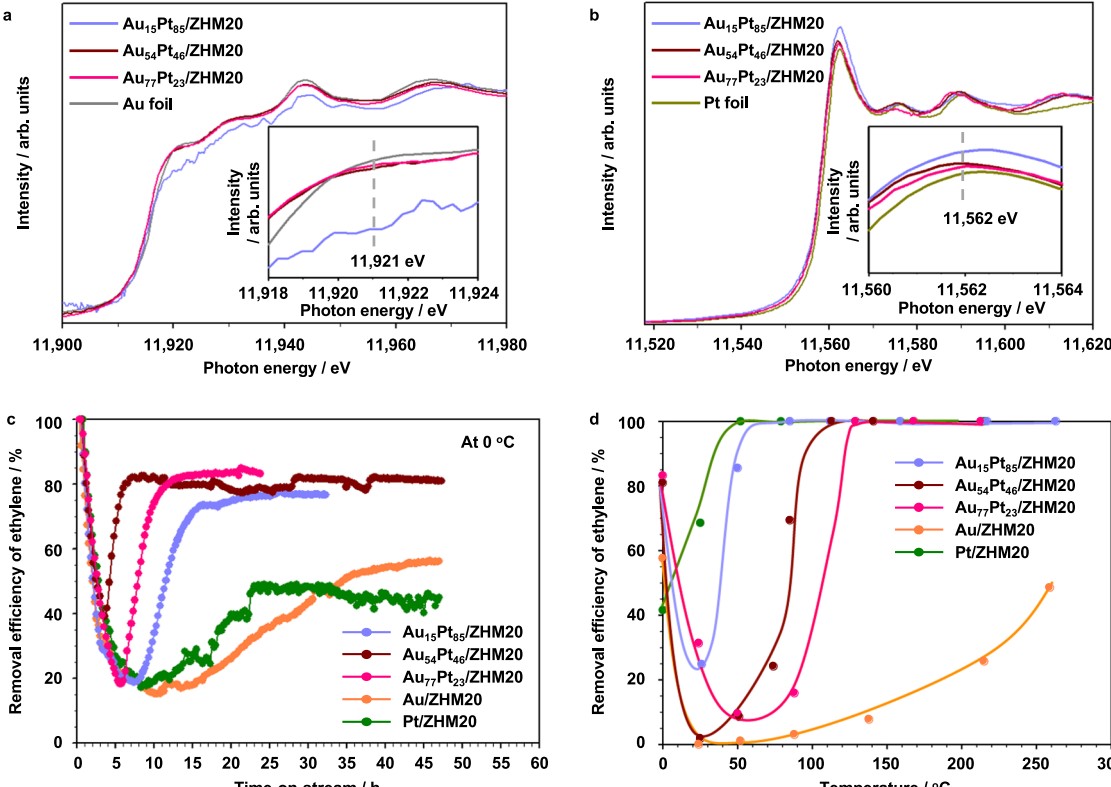

**Fig. 5 | Comparison with controls for C$_2$H$_4$ removal. a** Au $L_3$-edge and **b** Pt $L_3$-edge XANES spectra of Au-Pt/ZHM20 and Au foil/Pt foil. **c** C$_2$H$_4$ removal efficiencies of C$_2$H$_4$ with time-on-stream at 0 °C (Conditions: 50 ppm C$_2$H$_4$, 20% O$_2$ and N$_2$ balance; catalyst, 0.2 g; space velocity, 3000 mL h$^{-1}$ g$^{-1}$. **d** Temperature dependence of C$_2$H$_4$ removal efficiency over catalysts (Conditions: 50 ppm C$_2$H$_4$, 20% O$_2$ and N$_2$ balance; catalyst, 0.2 g; space velocity, 3000 mL h$^{-1}$ g$^{-1}$). Units arbitrary units.

## Methods

### Preparation of catalysts

The support of Mordenite 20 (ZHM20, SiO$_2$/Al$_2$O$_3$ of 18.3, particle size of 100–500 nm) was provided by the Catalysis Society of Japan (JRC-Z-HM20 (5) supplied by TOSOH Inc.). The sol immobilization method was used to prepare Pt/ZHM20, Au/ZHM20, and Au-Pt alloy NPs/ZHM20[14,35]. In details, poly(*N*-vinylpyrrolidone) (PVP, (C$_6$H$_9$NO)$_n$) with an average molecular weight of ca. 10 kDa (22.8 mg, K15, Tokyo Chemical Industry Co., LTD) was added to an aqueous solution (25 mL) that contained the desired molar ratio of H$_2$PtCl$_6$·6H$_2$O (Tanaka Kikinzoku Kogyo) and/or HAuCl$_4$·4H$_2$O (Tanaka Kikinzoku Kogyo). After cooling the mixture to 0 °C, 0.1 mol L$^{-1}$ NaBH$_4$ (5 mL, NaBH$_4$/metal (mol/mol) = 5, FUJIFILM Wako Pure Chemical Corporation) was added dropwise with vigorous stirring at 0 °C for 30 min to generate a colloid. Then ZHM20 (2 g) was added at room temperature followed by the addition of 1 vol% HCl (FUJIFILM Wako Pure Chemical Corporation) to adjust the mixture to pH 2.0. After vigorously stirring overnight at room temperature, the precipitate was filtered and thoroughly washed with deionized water (more than 3 L) until the pH of the filtrate solution was close to that of deionized water. The resulting solid was dried at 120 °C overnight and calcined at 500 °C for 2 h at a ramp rate of 5 °C min$^{-1}$ to obtain Pt/ZHM20, Au/ZHM20, and Au-Pt/ZHM20. For comparison, ZHM20 was calcined at 500 °C for 2 h before being used for the removal test.

### Materials characterization

The actual loading amounts of Pt and Au were measured by ICP-AES (Rigaku, Spectro Ciros CCD). The morphology and size distribution of Au NPs, Pt NPs, and Au-Pt alloy NPs were investigated by high-angle annular dark-field scanning transmission electron microscopy (HAADF-STEM, JEOL, JEM-3200FS). The crystalline feature of the Au-Pt alloy NPs was confirmed by high-resolution transmission electron

microscopy (HRTEM, JEOL, JEM-ARM200F NEOARM operating at 200 kV). The average size of NPs was calculated on the basis of at least 200 particles of each sample from different areas. Energy-dispersive X-ray spectroscopy (300 kV) was carried out to investigate the dispersion of Au and Pt atoms in the alloy NPs. The specific surface area and pore size distribution were determined from N$_2$ adsorption–desorption isotherms at 77 K on BELSORP-max (MicrotracBEL Japan). The sample was firstly pretreated at 300 °C under vacuum for 3 h before measurement. The pore size was analyzed from the desorption branch using a *t*-plot method. Considering the contribution of the micropore wall to the surface area, the surface area was also evaluated by the *t*-plot method[36]. The C$_2$H$_4$ adsorption isotherm was measured at 0 °C on BELSORP-max under ultrahigh vacuum and the samples were evacuated at 400 °C for 3 h before measurements.

Temperature-programmed desorption (TPD) measurement was carried out on an auto-chemisorption system (BELCAT-II, Japan) equipped with a mass spectrometer detector (BELMASS, MicrotracBEL, Japan). The sample (ca. 50 mg) for NH$_3$-TPD measurement was firstly pretreated at 250 °C for 1 h under He flow (30 mL min$^{-1}$). Then 5% NH$_3$/He was introduced at 100 °C for 30 min. The desorption profile of NH$_3$ was recorded from 100 °C to 750 °C under He with a flow rate of 30 mL min$^{-1}$. TPD measurement of the spent sample (at 0 °C) under He flow (30 mL min$^{-1}$) was carried out from 25 °C to 500 °C (5 °C min$^{-1}$) to investigate the possible products formed on the sample (0.1 g).

Pyridine adsorption was carried out by Fourier transformed infrared spectroscopy (FT-IR, FT/IR-6100, JASCO). The sample (ca. 10 mg) was pressed into a self-supported wafer of 10 mm in diameter and placed in the center of a horizontal-type heat chamber. After pretreatment at 500 °C for 90 min under 40 kPa O$_2$, the system was outgassed by vacuum, and 0.4 kPa of pyridine vapor was introduced for 30 min at room temperature. Then the temperature was increased to 150 °C for 60 min to remove the physical adsorption of pyridine.

The spectra of chemically adsorbed pyridine on the sample were then recorded. The spectra were recorded by accumulating 32 scans with a resolution of 4 cm$^{-1}$.

The powder X-ray diffraction (XRD) patterns of the samples were measured on a Rigaku Smartlab using Cu Kα1 radiation ($\lambda = 0.15406$ Å). Diffractions were recorded at a scan rate of 2° min$^{-1}$. X-ray absorption spectroscopy (XAS) measurement was carried out at the BL01B1 beamlines of SPring-8 (Hyogo, Japan) with the approval (proposal No. 2019B1386) of the Japan Synchrotron Radiation Research Institute (JASRI). Au $L_3$-edge and Pt $L_3$-edge X-ray absorption near edge structure (XANES) spectra were measured in fluorescence mode by using an Si double-crystal monochromator at room temperature. Athena software was used to analyze the obtained XANES spectra.

In-situ diffuse reflectance infrared Fourier transform spectroscopy of ethylene oxidation was carried out on a JASCO FT/IR-6100 spectrometer. All spectra were collected at a resolution of 4 cm$^{-1}$ after 64 scans. The sample was set in the heating chamber equipped with a diffuse reflectance accessory (ST Japan Heat Chamber HC-500) and a gas intake system. In each experiment, sample powder (6 mg) was placed in a DRIFT cell with a KBr window. The sample was firstly pretreated under N$_2$ flow (50 mL min$^{-1}$) at 250 °C for 1 h. After cooling the sample to 0 °C under N$_2$ flow, the background spectrum was taken. Due to the detection limit of FT-IR and considering fastening the adsorption of C$_2$H$_4$ on the support, a gas mixture of C$_2$H$_4$ (25 mL min$^{-1}$)/O$_2$ (20 mL min$^{-1}$)/N$_2$ (55 mL min$^{-1}$) was introduced to the sample at 0 °C for 30 min. Then the flow of C$_2$H$_4$ was stopped while maintaining the flow of O$_2$/N$_2$ for 5 min, and spectra were collected each minute.

## C$_2$H$_4$ removal tests

C$_2$H$_4$ removal measurements were performed on a stainless-steel fixed-bed flow reactor system which is connected to an online 490 Micro GC system (Agilent). The 490 Micro GC system is equipped with a micro thermal conductivity detector and PoraPLOT Q column. A catalyst (0.2 g) was set in a U-shaped glass reactor and pretreated at 150 °C for 2 h under N$_2$ flow (50 mL min$^{-1}$) before the test. Reactant gas that contained 50 ppm C$_2$H$_4$/20%O$_2$/N$_2$ balance was fed to the catalyst bed with a total flow rate of 10 mL min$^{-1}$, and the inlet and outlet concentrations of C$_2$H$_4$ and CO$_2$ were measured. We should note, according to the literature, that the experimental condition using a flow C$_2$H$_4$ with a concentration of 50 ppm has been commonly used for the evaluation of catalyst for C$_2$H$_4$ removal[6,7,26], including the commercially used catalyst developed by Prof. Fukuoka's group for the refrigerator units by Hitachi Global Life Solutions, Inc. (https://www.hitachi.com.au/products/product-categories/home-appliances/refrigerator/made-in-japan/RZX740KA.html). The space velocity was 3000 mL h$^{-1}$ g$^{-1}$ unless otherwise stated. The reaction temperature was controlled by using an ice bath for 0 °C, a water bath for temperatures from 0 °C to room temperature, and a furnace for temperatures higher than room temperature. C$_2$H$_4$ removal efficiency and CO$_2$ yield were calculated by Eqs. (1) and (2), respectively.

$$\text{C}_2\text{H}_4 \text{ removal efficiency} = \frac{[C_2H_4]_{in} - [C_2H_4]_{out}}{[C_2H_4]_{in}} \times 100 \quad (1)$$

$$\text{CO}_2 \text{ yield} = \frac{[CO_2]_{out}}{2 \times [C_2H_4]_{in}} \times 100 \quad (2)$$

## Data availability

The data generated in this study are provided in paper and Supplementary Information.

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

## Acknowledgements

This study was supported by the platform for technology and industry project of the Tokyo metropolitan government and by the Japan Society for the Promotion of Science (JSPS) Grant-in-aid for Scientific Research (22H00282 and 23H01763), "the Fundamental Research Funds for the Central Universities" and the National Key Research and Development Program of China (No. 2022YFC3703503). The authors are also grateful for the partial support of the JSPS and National Natural Science Foundation of China (NSFC) under the Joint Research Program (JRP with NSFC, No. JPJSJRP20191804), and Shanghai Municipal Bureau of Ecology and Environment (2022, HUHUANKE No. 30 and No. 33). XAS measurements were carried out at the BL01B1 beamlines of SPring-8 (Hyogo, Japan) with the approval (proposal No. 2019B1386) of the Japan Synchrotron Radiation Research Institute (JASRI). The authors also thank for the technical advice from NBC Meshtec Inc.

## Author contributions

T.M. and G.X. supervised the project. M.L. carried out the experiments and characterizations. H.W. prepared the catalysts. T.T., H.M., T.S., and Y.I. contributed to the data analyses. M.H., Y.L., and T.I. discussed the experimental results. J.H. conducted molecular dynamics simulations. M.L., T.M., and G.X. co-wrote the paper. All authors discussed the results and assisted during manuscript preparation.

## Competing interests

The authors declare no competing interests.
