## [Peer Review File · Nature Communications]

Selective formation of acetate intermediate prolongs robust ethylene removal at 0 °C for 15 daysREVIEWER COMMENTS

Reviewer #1 (Remarks to the Author):

The authors describe the synthesis and characterization of a catalyst (Au-Pt supported on mordenite 20) able to convert ethylene into acetic acid, which being solid at 0°C leaves the surface of the catalyst more exposed, resulting in an increased efficiency.

The idea itself is very interesting, and the work described scientifically sound, but there are certain points that require attention.

In general, the language needs to be properly checked and polished.

The title should be rewritten: it's not of immediate comprehension, and at the same time too narrow and too generic.

The authors provide three references (refs. 10-12) for the choice of the catalyst materials, but it would be very useful if they could motivate the choice not only of the metals but also of the support materials more in detail in the manuscript.

The authors execute the ethylene removal tests in flow, with an ethylene concentration of 50 ppm: are these conditions actually relevant for practical applications e.g. against vegetable spoilage? Moreover, the authors report the results in terms of percentual removal efficiency, but it would be very interesting to discuss more openly the rate of ethylene removal, and whether this is actually relevant for practical applications especially compared to existing solutions.

The reactivation of the spent catalyst requires a heat treatment at 450 °C for 2 h under nitrogen flow. These conditions probably serve to evaporate the acetic acid formed. However, the very high temperature and the necessity to operate in flow preclude the incorporation of the catalyst in e.g. food packaging materials. Maybe the authors could comment upon this point too.

Reviewer #2 (Remarks to the Author):

This paper presents a catalyst with impressive performance for removal of low levels of ethylene at 0 deg. C, which is relevant to reducing ethylene-induced ripening of fruits and vegetables during refrigerated storage and transport. This is interesting, and the performance is impressive. The work is potentially of more practical than scientific importance, as limited new insights are provided. The results and interpretation seem to be generally sound overall, but I have several questions and concerns that should be addressed before the paper is considered further.

The authors should better explain and justify their choices (of support, active metals, ethylene concentrations for testing and for operando studies, etc.) throughout the manuscript, and where necessary consider other options (e.g., testing ethylene removal at higher or lower feed concentration). Are the concentrations used those most relevant for a practical ethylene scrubber (like those that are commercially available, but presumably require power input and operate at higher temperature)?

The authors demonstrate effective removal over various periods of time, but perhaps a more relevant way of representing this data would be in terms of total quantity of ethylene removed. Does a catalyst that fails after 40 hours at 50 ppm ethylene fail after 20 hours at 100 ppm ethylene? Is its removal efficiency the same in both cases? Or is the relationship more complicated than that.

The authors do not adequately explain the origin of the minimum in all of the removal curves. While I understand the initial decrease, which follows the adsorption curve of the support, the observed shape required combining that with a long lag time for initiation of removal by the catalyst particles. Why would the removal curve not just transition smoothly from the initial adsorption to the steady-state value? Does a minimum amount of ethylene have to adsorb before reaction is initiated? Or is there some transport limitation? The authors could do more to elucidate this interesting effect, for example by looking at the dependence upon ethylene concentration and on flow rate. I note that the total amount of ethylene delivered in the removal experiments is small (about 1.3 micromoles per hour or 0.03 standard mL per hour) relative to the amount of catalyst present (about 10 micromoles metal).

The ethylene partial pressure used in DRIFTS measurements seems to be 5000 times higher, and the delivery rate 50,000 times higher, than in the ethylene removal experiments. The authors must better justify drawing conclusions from these experiments at much higher ethylene exposure and applying them to understand the behavior under ethylene removal conditions. Put another way, the amount of ethylene delivered in the 30 min exposure for drifts is the amount supplied in 25000 hours of ethylene removal experiments. The authors must also specify how much material was used in the DRIFTS study.

The authors should also present XRD patterns over a broad range.

The overlap between the Pt peak and support peak near 39 degrees makes determining much of anything from these difficult. The stretched horizontal scale of Fig. S2 (small range of diffraction angle) makes seeing shifts in peak position easier but may make the peaks themselves less clear to the reader. Are the other peaks from fcc Pt and Au evident over a broader range of diffraction angle, or are any impurity peaks present? Is the crystallite size from peak broadening in XRD comparable to the particle size?

I was surprised to see a significant increase in surface area upon addition of a small amount of active metal to the support.

Do the authors have an explanation for this increase? Was the bare support calcined before measuring its surface area? The particles are much larger than the pore sizes of mordenite. Are the metal particles on the exterior surface, or embedded? What was the particle size of the mordenite?

The pure Pt particles are much smaller than the pure Au, and the deposition efficiency (ratio of actual content to intended content) is lower for Pt. As expected, the particle size decreases with increasing Pt content in the alloys. However, based on the size distribution in Fig. S1 and S11, even a small amount of Pt dramatically decreases the particle size. Size increases only about 60% from 0% Au to 85% Au, then by almost a factor of 3 from 85% Au to 100% Au. Is there a simple explanation for this?

Adding bright-field TEM/HRTEM instead of only dark-field may provide the reader with additional insight into the particle structure (for example, phase or elemental segregation may show up there). HRTEM would also provide insights into the degree of crystallinity of the metal, which XRD suggests is low.

Manuscript ID: NCOMMS-22-47650

Updated title: "Selective formation of acetate intermediate prolongs robust ethylene removal at 0 °C for 15 days"

Reviewer #1:

The authors describe the synthesis and characterization of a catalyst (Au-Pt supported on mordenite 20) able to convert ethylene into acetic acid, which being solid at 0 °C leaves the surface of the catalyst more exposed, resulting in an increased efficiency.

The idea itself is very interesting, and the work described scientifically sound, but there are certain points that require attention.

Response:

We appreciate the reviewer's positive feedback and show our actions point-by-point below.

Q1: *In general, the language needs to be properly checked and polished.*

Response:

We have carefully checked and polished the manuscript throughout.

Q2: *The title should be rewritten: it's not of immediate comprehension, and at the same time too narrow and too generic.*

Response:

As suggested by the reviewer, we now change the title to "Selective formation of acetate intermediate prolongs robust ethylene removal at 0 °C for 15 days".

Q3: *The authors provide three references (refs. 10-12) for the choice of the catalyst materials, but it would be very useful if they could motivate the choice not only of the metals but also of the support materials more in detail in the manuscript.*

Response:

We have now explained the choices of the catalysts and support materials in **page 4** of the

revised manuscript as discussed below, and we have cited 9 references with the reference numbers of **#13 to #21** in the revised manuscript.

In **page 4**: We should note that the choices of both catalysts and support materials are critical for C₂H₄ removal. Alloy nanoparticles (NPs) are important catalysts for the conversion of hydrocarbons¹³ due to the different electronic states and structures compared to the monometallic NPs¹⁴. For example, the dominant adsorption mode of C₂H₄ on a Pt(111) surface is the di-σ adsorbed mode up to ~250 K¹⁵⁻¹⁷, and it would change to a stable phase of ethylidyne species (CH₃C) as the temperature increased^{18,19}. Introduction of other metals would result in geometric and electronic modulations on Pt and therefore optimize the key adsorption species on Pt, such as the addition of Sn would suppress the formation of ethylidyne²⁰ and the addition of Au would enhance the interaction between Pt and ethylidyne species²¹. In addition, C₂H₄ can adsorb on Au sites at 263 K and 203 K with a lower heat of adsorption than that on Pt sites²¹. Therefore, we expect that the Au-Pt alloy could be a good catalyst candidate for the adsorption and transformation of C₂H₄ at low temperatures. For the support materials, given that Brønsted acidity promotes the adsorption of C₂H₄¹⁰⁻¹², we select Mordenite 20 (denoted as ZHM20) as the acidic support for Au-Pt alloy NPs due to its abundant acid sites, especially Brønsted acid sites, and large surface area⁹.

Q4: The authors execute the ethylene removal tests in flow, with an ethylene concentration of 50 ppm: are these conditions actually relevant for practical applications e.g. against vegetable spoilage?

Response:

Thanks for the reviewer's comment. According to the literature, the experimental condition using a flow C₂H₄ with a concentration of 50 ppm has been commonly used for the evaluation of catalysts for C₂H₄ removal (ACS Catal., 2020, 10, 13257-13268; ACS Sustainable Chem. Eng., 2018, 6, 11480-11486; Chem. Lett., 2018, 47, 1000-1002; Angew. Chem. Int. Ed., 2013, 52, 6265-6268). This condition also applies to the commercially used catalysts, for example, the catalyst developed by Prof. Fukuoka's group for the refrigerator units by Hitachi Global Life Solutions, Inc. (<https://www.hitachi.com.au/products/product-categories/home-appliances/refrigerator/made-in-japan/RZX740KA.html>). This catalyst has been evaluated based on the similar experimental condition during development.

Regarding the concerns for the C₂H₄ concentrations in practical applications, we agree that the 50 ppm C₂H₄ is higher than the actual concentration, which depends on the amount and types of fruits and vegetables, as well as the storage space. We noted that the

threshold whether the ripening of the perishables proceeds or not is only 0.1 ppm (*Chem. Rev.*, 2013, 113, 5029-5070). However, we should also note that this low concentration is difficult, experimentally, to precisely introduce into the reactor via mass flow controller. And the corresponding unreactive C₂H₄ and possible products can be also hardly detected by gas chromatography.

Therefore, we used 50 ppm C₂H₄ as the evaluation experimental condition in this study. We have added the above discussions to **page 16** in the revised manuscript.

Q5: *Moreover, the authors report the results in terms of percentual removal efficiency, but it would be very interesting to discuss more openly the rate of ethylene removal, and whether this is actually relevant for practical applications especially compared to existing solutions.*

Response:

We thank these valuable suggestions from the reviewer. We now calculate the C₂H₄ removal rate on Au₅₄Pt₄₆/ZHM20 in the steady state at 0 °C to be 120 mL_(ethylene)/kg·h, which is about 5x higher than the reported commercially used Pt/SBA-15 of 25 mL_(ethylene)/kg·h (*ACS Sustainable Chem. Eng.*, 2018, 6, 11480-11486). This rate is also much higher than that of C₂H₄ generated by fruits, such as apple (0.28 mL_(ethylene)/kg·h) according to the semi-practical conditions for the preservation of perishables (*Catal. Sci. Technol.*, 2022, 12, 3116-3122), suggesting the promising actual application of Au₅₄Pt₄₆/ZHM20 catalyst.

In addition, compare this catalytic process for eliminating C₂H₄ with other existing solutions, we noticed that most of the traditional C₂H₄ removal methods have significant shortcomings. For example, adsorbents such as activated carbon cannot be used for a long time due to the limited adsorption capacity; chemical oxidants are toxic and contain potential safety hazards during long-term use; photocatalytic technology requires high equipment costs because of the need for ultraviolet light sources. Therefore, the catalytic process, especially when we use a catalyst with robust activity and stability such as Au₅₄Pt₄₆/ZHM20 produced in this work, would provide new opportunities for removing trace ethylene for a long time at low temperatures.

We have provided the related descriptions on **pages 6-7** and **page 12** in the revised manuscript.

Q6: *The reactivation of the spent catalyst requires a heat treatment at 450 °C for 2 h under*

nitrogen flow. These conditions probably serve to evaporate the acetic acid formed. However, the very high temperature and the necessity to operate in flow preclude the incorporation of the catalyst in e.g. food packaging materials. Maybe the authors could comment upon this point too.

Response:

We totally agree with the comment from the reviewer on the difficulty to incorporate this catalyst into food packaging materials. However, we expect the usage of this catalyst in a box-like device with air flow system, which will locate in the space for the cold-chain storage and transportation. The spent catalyst will be collected and heat-treated for recycle-use. We also tried to decrease the temperature of the heat treatment to 150 °C that some food packaging materials could be maintained. Unfortunately, the C₂H₄ removal efficiency cannot be recovered, showing a starting 65% efficiency and decreasing to below 5% in several hours (**Figure R1**). However, we anticipate that the catalyst could be still possible for involving in food packaging materials, such as disposable packaging ones (and recyclable), when we consider the long-term stability for around 15 days of this catalyst for continuous C₂H₄ removal. We have commented this in **page 7** of the revised manuscript.

Figure R1 | Time courses for C₂H₄ removal over Au₅₄Pt₄₆/ZHM20 at 0 °C. Heat treatment was conducted at 150 °C for 2 h under N₂ flow (50 mL min⁻¹).

Reviewer #2

This paper presents a catalyst with impressive performance for removal of low levels of ethylene at 0 deg. C, which is relevant to reducing ethylene-induced ripening of fruits and vegetables during refrigerated storage and transport. This is interesting, and the performance is impressive. The work is potentially of more practical than scientific importance, as limited new insights are provided. The results and interpretation seem to be generally sound overall, but I have several questions and concerns that should be addressed before the paper is considered further.

Response:

We thank the reviewer for the constructive comments, and we have revised the manuscript accordingly as detailed below.

Q1: *The authors should better explain and justify their choices (of support, active metals, ethylene concentrations for testing and for operando studies, etc.) throughout the manuscript, and where necessary consider other options (e.g., testing ethylene removal at higher or lower feed concentration). Are the concentrations used those most relevant for a practical ethylene scrubber (like those that are commercially available, but presumably require power input and operate at higher temperature)?*

Response:

We now summarize the concerns from the reviewer and repones them point-by-point as followings.

(I) The choices of support and active metals.

We now provide the explanation of the choices of the support and active metals in **page 4** of the revised manuscript that reads: *“We should note that the choices of both catalysts and support materials are critical for C₂H₄ removal. Alloy nanoparticles (NPs) are important catalysts for the conversion of hydrocarbons¹³ due to the different electronic states and structures compared to the monometallic NPs¹⁴. For example, the dominant adsorption mode of C₂H₄ on a Pt(111) surface is the di-σ adsorbed mode up to ~250 K¹⁵⁻¹⁷, and it would change to a stable phase of ethylidyne species (CH₃C) as the temperature increased^{18,19}. Introduction of other metals would result in geometric and electronic modulations on Pt and therefore optimize the key adsorption species on Pt, such as the addition of Sn would suppress the formation of ethylidyne²⁰ and the addition of Au would enhance the interaction between Pt and ethylidyne species²¹. In addition, C₂H₄ can adsorb*

on Au sites at 263 K and 203 K with a lower heat of adsorption than that on Pt sites²¹. Therefore, we expect that the Au-Pt alloy could be a good catalyst candidate for the adsorption and transformation of C₂H₄ at low temperatures. For the support materials, given that Brønsted acidity promotes the adsorption of C₂H₄¹⁰⁻¹², we select Mordenite 20 (denoted as ZHM20) as the acidic support for Au-Pt alloy NPs due to its abundant acid sites, especially Brønsted acid sites, and large surface area⁹.”

(II) The choice of C₂H₄ concentrations for testing.

According to the literature, the experimental condition using a flow C₂H₄ with a concentration of 50 ppm has been commonly used for the evaluation of catalyst for C₂H₄ removal (*ACS Catal.*, 2020, 10, 13257-13268; *ACS Sustainable Chem. Eng.*, 2018, 6, 11480-11486; *Chem. Lett.*, 2018, 47, 1000-1002; *Angew. Chem. Int. Ed.*, 2013, 52, 6265-6268). This condition also applies to the commercially used catalysts, for example, the catalyst (Pt/SBA-15) developed by Prof. Fukuoka's group for the refrigerator units by Hitachi Global Life Solutions, Inc. (<https://www.hitachi.com.au/products/product-categories/home-appliances/refrigerator/made-in-japan/RZX740KA.html>). The catalysts have been evaluated based on the similar experimental conditions during development. Therefore, we used 50 ppm C₂H₄ for testing in this study. We have added the related discussions of this part to **page 16** in the revised manuscript.

(III) The choice of C₂H₄ concentrations for *operando* studies.

Just for clarity, conducting the *operando* measurements same as that used in testing condition is difficult because 50 ppm C₂H₄ in DRIFT analysis is negligible. In addition, we found the adsorption of C₂H₄ on catalyst in the first stage, therefore, we chose 5000 times higher concentration of C₂H₄ and 10 times higher flow rate to make sure the adsorption on support is complete before collecting the results related to the reaction processes, such as the information about the *in-situ* formed species at 0 °C. We have now explained this in **page 15** in the revised manuscript that reads: “Due to the detection limit of FT-IR and considering about fastening the adsorption of C₂H₄ on the support, a gas mixture of C₂H₄ (25 mL min⁻¹)/O₂ (20 mL min⁻¹)/N₂ (55 mL min⁻¹) was introduced to the sample at 0 °C for 30 min.”

(IV) The concentrations used in this work compared to a practical C₂H₄ scrubber.

As discussed above, Prof. Fukuoka's group has developed a commercially used Pt/SBA-15 catalyst for the refrigerator units by Hitachi Global Life Solutions, Inc. We have calculated the C₂H₄ removal rate on Au₅₄Pt₄₆/ZHM20 in the steady state at 0 °C to be 120 mL_(ethylene)/kg·h, which is ~5-fold higher than the Pt/SBA-15 of 25 mL_(ethylene)/kg·h (*ACS Sustainable Chem. Eng.*, 2018, 6, 11480-11486). This rate is also much higher than that of

C₂H₄ generated by fruit, such as apple (0.28 mL_(ethylene)/kg·h) according to the semi-practical conditions for the preservation of perishables (*Catal. Sci. Technol.*, 2022, 12, 3116-3122), suggesting the promising actual application. In addition, compare this catalytic process for eliminating C₂H₄ with other existing practical ways, we noticed that most of the traditional C₂H₄ removal methods have shortages. For example, adsorbents such as activated carbon cannot be used for a long time due to the limited adsorption capacity; chemical oxidants are toxic and contain potential safety hazards during long-term use; photocatalytic technology requires high equipment costs owing to the need for ultraviolet light sources. Therefore, the catalytic process, especially when we use a catalyst with robust activity and stability such as Au₅₄Pt₄₆/ZHM20 produced in this work, would provide new opportunities for removing trace C₂H₄ for a long time at low temperatures. We have now described accordingly on **pages 6-7** and **page 12** in the revised manuscript.

Q2: *The authors demonstrate effective removal over various periods of time, but perhaps a more relevant way of representing this data would be in terms of total quantity of ethylene removed. Does a catalyst that fails after 40 hours at 50 ppm ethylene fail after 20 hours at 100 ppm ethylene? Is its removal efficiency the same in both cases? Or is the relationship more complicated than that.*

Response:

As suggested by the reviewer, we have now presented the total quantity of C₂H₄ removed on each catalyst prepared in this work, as well as those of recent reported catalysts in literature, as shown in **Table R1** (Updated **Supplementary Table 3** in the revised SI). Due to the C₂H₄ removal tests in references do not proceed to the point that the catalysts are fully deactivated, we therefore can only calculate the quantity of ethylene removed based on the provided curves in literature.

We noted that the reported catalysts show a low total C₂H₄ removal quantity with a short operation time, which are impractical for the long-term and long-distance cold-chain preservation and storage of fruits and vegetables. Therefore, we conclude that the total C₂H₄ removal quantity of 4.4 mL combined with the ultra-long operation time of 15 days of Au₅₄Pt₄₆/ZHM20 catalyst are both critical, especially when we considerate the two orders of magnitude lower C₂H₄ generation rate by fruits and vegetables (such as 0.28 mL_(ethylene)/kg·h for apple) than that of Au₅₄Pt₄₆/ZHM20 catalyst (120 mL_(ethylene)/kg·h), in which the catalyst is capacity for eliminating these trace amounts of C₂H₄ in cold-chain environment.

We have added the above discussions to **pages 6-7** in the revised manuscript.

Regarding the concerns of the stability of the catalysts at different C₂H₄ concentrations, since the reviewer has a similar concern in the next comment (Q3), we will response it in details in the next question below.

Table R1 (Updated Supplementary Table 3) | Comparisons of the C₂H₄ removal efficiency and stability over Au₅₄Pt₄₆/ZHM20 with recent reports in literature.

Catalyst	Temperature / °C	C ₂ H ₄ removal efficiency / %	Time-on-stream / h	Total removal ^[a] / mL	Reference
Au ₅₄ Pt ₄₆ /ZHM20	0	100 → 2	359	4.40	This study
WTi-Pt2.5	0	100 → 38	2.5 ^[b]	0.096	ACS Nano , 13 , 14337-14347 (2019)
Pt/SBA-15	0	100 → 45	5 ^[b]	0.094	ACS Catal. , 10 , 13257-13268 (2020)
Pt/MCM-41	0	99.8 → 55	2 ^[b]	0.049	Angew. Chem. Int. Ed. , 52 , 6265-6268 (2013)
Au/Co ₃ O ₄	0	100 → 98	1 ^[b]	0.001	Environ. Sci. Technol. , 42 , 8947-8951 (2008)
Ag/Beta	25 ^[c]	100 → 0	24	2.58	ACS Catal. , 8 , 1248-1258 (2018)
Pt/F-ZSM-5	25 ^[c]	100 → 98	11 ^[b]	1.66	Catal. Sci. Technol. , 8 , 1988-1996 (2018)

^[a] The total C₂H₄ removal amount is calculated from the removal curve. ^[b] Deactivation time was not shown in literature. ^[c] No C₂H₄ removal results were conducted at around 0 °C in literature.

Q3: *The authors do not adequately explain the origin of the minimum in all of the removal curves. While I understand the initial decrease, which follows the adsorption curve of the support, the observed shape required combining that with a long lag time for initiation of removal by the catalyst particles. Why would the removal curve not just transition smoothly from the initial adsorption to the steady-state value? Does a minimum amount of ethylene have to adsorb before reaction is initiated? Or is there some transport limitation? The authors could do more to elucidate this interesting effect, for example by looking at the dependence upon ethylene concentration and on flow rate. I note that the total amount of ethylene delivered in the removal experiments is small (about 1.3 micromoles per hour or 0.03 standard mL per hour) relative to the amount of catalyst present (about 10 micromoles metal).*

Response:

Following the reviewer's suggestions, we went back to lab and conducted the tests with different C₂H₄ concentrations (25 and 100 ppm C₂H₄ at 10 mL min⁻¹ flow rate). Just for clarity, the Au₅₄Pt₄₆/ZHM20 catalyst we used here is the one in the original manuscript that has been stored for two years, in order to further demonstrate its durability for C₂H₄ removal. We firstly heat treated this catalyst at 450 °C for 2 hours before tests. As illustrated in **Figure R2**, we found that the conversion efficiency of C₂H₄ removal can still achieve 75% after aging the catalyst for two years, suggesting the excellent stability of the catalyst, which is critical for the actual applications. In addition, we observed that all removal tests on Au₅₄Pt₄₆/ZHM20 catalyst exhibit a curve with U-shape. As commented by the reviewer, we agree that the initial decrease follows the adsorption of C₂H₄ on the support of ZHM20. We also agree with the point that the catalyst needs to adsorb a minimum amount of C₂H₄ before reaction is initiated. This could be because the support contains abundant acid sites, especially Brønsted acid sites that may more favor the C₂H₄ adsorption than Au-Pt alloys. Therefore, the catalytic reaction for selectively converting C₂H₄ to acetic acid could not be started owing to the lack of C₂H₄ reactant on Au-Pt alloyed catalyst, since most of C₂H₄ molecules are trapped by the ZHM20 support in the initial hours.

Figure R2 (New Supplementary Fig. 10 in the revised SI) | C₂H₄ removal efficiencies with time-on-stream over Au₅₄Pt₄₆/ZHM20 at 0 °C with different C₂H₄ concentrations (reaction condition: 25, 50, or 100 ppm C₂H₄, 20% O₂ and N₂ balance; catalyst, 0.2 g).

We also noted some interesting phenomena in **Figure R2** that may help us to understand the detailed reaction processes. And we summarize and discuss them as followings.

- (i) One can see, with the higher concentration of C₂H₄ we applied, the quicker decrease of C₂H₄ removal efficiency in the first several hours. This could reflect the adsorption curve of the ZHM20 support (**Figure R2-I** below).

Figure R2-I | The initial decreasing parts for C₂H₄ removal in Figure R2.

- (ii) When the curve reaches the minimum point, the catalytic reaction for converting C₂H₄ may start and the removal efficiency increases again, thus showing a U-shape curve.
- (iii) We noted, with a low C₂H₄ concentration of 25 ppm, that the catalyst shows a delay activation than that of 50 ppm C₂H₄, suggesting the transport limitation under the condition of 25 ppm C₂H₄, in which the catalytic reaction is slower than that of 50 ppm C₂H₄.
- (iv) The similar removal efficiency in the steady state under conditions of 25 and 50 ppm C₂H₄ suggests the catalyst may have a maximum removal efficiency of ~80% under the flow rate of 10 mL min⁻¹. This may also be due to the transport limitation. However, we are unable to do tests of flow rates lower than 10 mL min⁻¹ owing to the limitation of the mass flow controller.
- (v) We further found, in the high C₂H₄ concentration case (100 ppm C₂H₄), that the removal efficiency is about half of that of 50 ppm C₂H₄ (**Figure R2**). This suggests a similar catalytic rate in both conditions (**Figure R2-II** below).

Figure R2-II | C₂H₄ removal rates of the Au₅₄Pt₄₆/ZHM20 catalyst in the steady state (calculated from the data in Figure R2) with different C₂H₄ concentrations.

To further understand the origin of the minimum in the removal curves, we then increased the flow rate in C₂H₄ removal experiments to 20 mL min⁻¹, in which the space velocity is 6000 mL h⁻¹ g⁻¹. As we can observe in **Figure R3**, with the higher flow rate we set, the quicker decrease of C₂H₄ removal efficiency is observed in the first several hours. This could reflect the adsorption curve of the ZHM20 support. Additionally, we found the C₂H₄ removal rates are similar at both flow rates, indicating the flow rate may have negligible influence on the activity of Au₅₄Pt₄₆/ZHM20 catalyst (**Figure R3-I** below).

Figure R3 (New Supplementary Fig. 11 in the revised SI) | C₂H₄ removal efficiencies with time-on-stream over Au₅₄Pt₄₆/ZHM20 at 0 °C with different flow rates (reaction condition: 50 ppm C₂H₄, 20% O₂ and N₂ balance; catalyst, 0.2 g; space velocity, 3000 or 6000 mL h⁻¹ g⁻¹).

Figure R3-I | C₂H₄ removal rates of the Au₅₄Pt₄₆/ZHM20 catalyst in the steady state (calculated from the data in Figure R3) with different flow rates.

Based on the above discussion, we noted, at a low C₂H₄ concentration of 25 ppm, that the catalyst exhibits a delay activation and a similar C₂H₄ removal efficiency compared to those of 50 ppm, suggesting the transport limitation under the condition of 25 ppm C₂H₄. However, when we increased the C₂H₄ concentration to 50 ppm or higher 100 ppm, the C₂H₄

concentrations and flow rates may have negligible influence on the C₂H₄ removal activity of Au₅₄Pt₄₆/ZHM20 catalyst.

Moreover, regarding the concerns about the stability of the catalyst that the reviewer commented in Q2, we confirmed that the conversion efficiency of catalyst does gradually decrease after 20 hours in steady state at 100 ppm C₂H₄ (**Figure R2**), indicating the selective formation of solid-like acetic acid that would gradually cover the catalyst surface.

We have now discussed accordingly in **page 6** and **page 10** in the revised manuscript, and added new **Supplementary Note 1** and **Figs. 10 and 11** in the revised SI.

Q4: *The ethylene partial pressure used in DRIFTS measurements seems to be 5000 times higher, and the delivery rate 50,000 times higher, than in the ethylene removal experiments. The authors must better justify drawing conclusions from these experiments at might higher ethylene exposure and applying them to understand the behavior under ethylene removal conditions. Put another way, the amount of ethylene delivered in the 30 min exposure for DRIFTS is the amount supplied in 25000 hours of ethylene removal experiments. The authors must also specify how much material was used in the DRIFTS study.*

Response:

The mass amount of catalysts used in DRIFT study is 6 mg, which we have now specified in **page 15** in the revised manuscript.

Regarding the concerns of the different conditions in DRIFT measurements and C₂H₄ removal experiments, as we responded to the previous comment Q1, 50 ppm C₂H₄ is negligible if we use it in DRIFT measurements. In addition, we found the adsorption of C₂H₄ on catalyst in the first stage, therefore, we chose higher C₂H₄ concentration and flow rate to make sure the adsorption on support is complete before collecting the results related to the reaction processes, such as the information about the *in-situ* formed species at 0 °C.

We now show the related explanation in **page 15** in the revised manuscript that reads: “*In each experiment, sample powder (6 mg) was placed in a DRIFT cell with a KBr window. ... Due to the detection limit of FT-IR and considering about fastening the adsorption of C₂H₄ on the support, a gas mixture of C₂H₄ (25 mL min⁻¹)/O₂ (20 mL min⁻¹)/N₂ (55 mL min⁻¹) was introduced to the sample at 0 °C for 30 min.*”

Q5: *The authors should also present XRD patterns over a broad range. The overlap between the Pt peak and support peak near 39 degrees makes determining much of*

anything from these difficult. The stretched horizontal scale of Fig. S2 (small range of diffraction angle) makes seeing shifts in peak position easier but may make the peaks themselves less clear to the reader. Are the other peaks from fcc Pt and Au evident over a broader range of diffraction angle, or are any impurity peaks present? Is the crystallite size from peak broadening in XRD comparable to the particle size?

Response:

As commented by the reviewer, we now show the broad range (10~70°) of XRD patterns of the prepared catalysts in **Figure R4** (Updated **Supplementary Fig. 2** in the revised SI). The support of ZHM20 shows typical diffraction peaks corresponding to PDF No. 98841. Negligible structural change can be observed after the deposition of Au, Pt, or Au₅₄Pt₄₆ nanoparticles (NPs). However, we found some diffraction peaks between 2θ = 37.5° and 40.5°, which can be associated to the Pt(111), Au(111), and Au-Pt alloy. The inset enlarged image presents patterns with a range of 37.5° to 40.5°, showing that the broad peak centered at 38.8° can be assigned to the Au-Pt alloy NPs. We should also note that the low intensity of the peaks for metal or alloy may be unsuitable to confirm the crystallite size or particle size, because the metal loading amount is only 1 wt%, which is close to the detection limit of XRD.

Figure R4 (Updated Supplementary Fig. 2) | XRD patterns of ZHM20, Au/ZHM20, Au₅₄Pt₄₆/ZHM20, and Pt/ZHM20.

Q6: I was surprised to see a significant increase in surface area upon addition of a small amount of active metal to the support. Do the authors have an explanation for this increase? Was the bare support calcined before measuring its surface area? The particles are much

larger than the pore sizes of mordenite. Are the metal particles on the exterior surface, or embedded? What was the particle size of the mordenite?

Response:

Just for clarity, the bare support was calcined at 500 °C before measuring the surface area. As commented by the reviewer, we have redone the surface area test of ZHM20, which is 835 m²/g, the highest compared to the loaded catalysts. The results have been updated in **Table R2** (Updated **Supplementary Table 2** in the revised SI) and the corresponding description has been modified in **page 5** in the revised manuscript that reads: “We further calculated surface areas of Au₅₄Pt₄₆/ZHM20 together with other two controls and bare support ZHM20 (calcined at 500 °C) to be from 766 m² g⁻¹ to 835 m² g⁻¹ by nitrogen adsorption and desorption isotherms.”

For the concerns of the deposition position of the metal particles, since the inner pore size of ZHM20 is only 0.58 nm, much smaller than the sizes of metal nanoparticles, we thus conclude that the metal particles are deposited on the exterior surface of the support. And the particle size of the mordenite (provided by the Catalysis Society of Japan (JRC-Z-HM20 (5) supplied by TOSOH Inc.) is 100~500 nm. We have updated these details in **page 5** and **page 13** in the revised manuscript and in **Supplementary Table 2** in the revised SI.

Table R2 (Updated Supplementary Table 2) | Physical properties of ZHM20, Au/ZHM20, Pt/ZHM20, and Au-Pt/ZHM20.

Sample	Particle size ^[a] (nm)	Surface area ^[b] (m ² /g)	Pore size ^[b] (nm)	Acid amount ^[c] (mmol/g)	Ratio of Brønsted/Lewis ^[d]
ZHM20-AC500	N.A.	835	0.58	1.0	8.7
Au/ZHM20	22 ± 15	766	0.58	1.1	N.A.
Pt/ZHM20	5.0 ± 1.2	828	0.58	0.98	N.A.
Au ₅₄ Pt ₄₆ /ZHM20	6.5 ± 2.1	826	0.58	0.96	6.0
Au ₁₅ Pt ₈₅ /ZHM20	5.8 ± 2.0	815	0.58	1.0	5.9
Au ₇₇ Pt ₂₃ /ZHM20	8.4 ± 2.8	802	0.58	0.97	6.4

^[a] Determined by HAADF-STEM. The particle size presents the metal nanoparticles. ^[b] Calculated by the *t*-plot method. ^[c] Measured and calculated from NH₃-TPD. ^[d] Calculated from pyridine adsorption. N.A., not applicable.

Q7: *The pure Pt particles are much smaller than the pure Au, and the deposition efficiency (ratio of actual content to intended content) is lower for Pt. As expected, the particle size decreases with increasing Pt content in the alloys. However, based on the size distribution in Fig. S1 and S11, even a small amount of Pt dramatically decreases the particle size. Size increases only about 60% from 0% Au to 85% Au, then by almost a factor of 3 from 85% Au to 100% Au. Is there a simple explanation for this?*

Response:

We appreciate this reviewer's comment. This is because, in this work, we used sol immobilization method to deposit the Au-Pt alloy nanoparticles by using PVP as the protecting ligand. This method can deposit Pt nanoparticles and Au-Pt alloy nanoparticles with small sizes on ZHM20 (*J. Catal.*, 2021, 402, 101-113); however, it is not suitable to deposit Au nanoparticles (*Chem. Rev.*, 2019, 120, 464-525). Moreover, fully removing the PVP ligand requires a temperature higher than 500 °C (*Catal. Today*, 2023, 410, 143-149), and such high calcination temperature will further cause the aggregation of Au nanoparticles, resulting the large size of 22 ± 15 nm. We have provided the explanation of this part in **page 5** in the revised manuscript.

Q8: *Adding bright-field TEM/HRTEM instead of only dark-field may provide the reader with additional insight into the particle structure (for example, phase or elemental segregation may show up there). HRTEM would also provide insights into the degree of crystallinity of the metal, which XRD suggests is low.*

Response:

We have now collected the HRTEM images of the Au-Pt alloy loaded catalysts. As shown in **Figure R5**, we can see that all three Au-Pt alloys have clear fringe spacings, indicating the high crystalline feature of these metal alloys. Moreover, we detected the elemental mappings of Au-Pt/ZHM20 controls (**Figures R6 and R7**), in which show the well dispersed Au and Pt elements in NPs in each catalyst. We have included these results in **pages 10-11** in the revised manuscript and **Supplementary Figs. 14-16** in the revised SI.

Figure R5 (New Supplementary Fig. 14) | HRTEM images of (a) Au₁₅Pt₈₅/ZHM20, (b) Au₅₄Pt₄₆/ZHM20, and (c) Au₇₇Pt₂₃/ZHM20.

Figure R6 (New Supplementary Fig. 15) | HAADF-STEM images and corresponding elemental mappings of (a) Au₁₅Pt₈₅/ZHM20 and (b) Au₇₇Pt₂₃/ZHM20. Scale bar, 10 nm.

Figure R7 (New Supplementary Fig. 16) | HAADF-STEM images and corresponding liner analyses of Pt and Au elementals in (a) $\text{Au}_{15}\text{Pt}_{85}/\text{ZHM20}$, (b) $\text{Au}_{54}\text{Pt}_{46}/\text{ZHM20}$, and (c) $\text{Au}_{77}\text{Pt}_{23}/\text{ZHM20}$.

REVIEWERS' COMMENTS

Reviewer #2 (Remarks to the Author):

In their revision, the authors have adequately addressed the reviewer concerns, and I recommend acceptance of the revised manuscript.

Manuscript ID: NCOMMS-22-47650A

Title: "Selective formation of acetate intermediate prolongs robust ethylene removal at 0 °C for 15 days"

Reviewer #2:

In their revision, the authors have adequately addressed the reviewer concerns, and I recommend acceptance of the revised manuscript.

Response:

We appreciate the reviewer's positive feedback and recommendation.